# Foundation Research on Physicochemical Properties of Mine Insulation Materials

**Chuanbin Hou [1], Song Xin [1,2,*], Long Zhang [1], Shangxiao Liu [1] and Xiao Zhang [1]**

[1] College of Mining and Safety Engineering, Shandong University of Science and Technology, Qingdao 266590, China; chuanbinhou@163.com (C.H.); aghlnoz@163.com (L.Z.); liushangxiao_24@163.com (S.L.); zhangxiao19961996@163.com (X.Z.)

[2] Mine Disaster Prevention and Control-Ministry of State Key Laboratory Breeding Base, Shandong University of Science and Technology, Qingdao 266590, China

* Correspondence: xinsong@sdust.edu.cn; Tel.: +86-0532-8605-7969

**Abstract:** The known cooling methods for the high-temperature operating environment of a mine mainly include ventilation, refrigeration, heat insulation, and individual protection. Among them, the superior performance and wide application of the heat insulation materials have attracted the attention of the coal mining industry. In this paper, three types of mineral insulation materials were prepared using basalt fiber, glass fiber, vitrified microbeads in combination with cement, sand, high-strength ceramsite, water, etc. In addition, the thermal conductivity and compressive strength of the prepared specimens were assessed. The results show that the test specimen containing basalt fiber had a great thermal insulation effect and achieved the required compressive strength. Furthermore, according to the COMSOL simulation results, the test specimen containing basalt fiber had a better thermal insulation effect than the ordinary concrete materials. Therefore, the research results of this article have guiding significance to search for new mine thermal insulation materials.

**Keywords:** mine cooling; insulation materials; basalt fiber; thermal conductivity; compressive strength; COMSOL

## 1. Introduction

As the main energy source of China, coal has made great contributions to the long-term development of industry. The energy structure of "rich coal, lean oil and low gas" in China will not change for a long time. It is predicted that China's coal demand will be 3.9–4.4 billion tons, 4.5–5.1 billion tons, and 3.8 billion tons in 2020, 2030, and 2050, respectively [1]. As shallow resources get depleted and energy demand keeps increasing, deep mining is becoming normal. Heping Xie, an academician of Engineering Academy in China, pointed out that "high geostress, high ground temperature and high osmotic pressure" are the main bottlenecks affecting deep mining in the future [2]. Among them, the thermal damage in mines caused by high ground temperature has brought problems to coal mine workers. High ground temperature is the main cause of thermal damage to the mine. In addition to high ground temperature, other factors, including air compression heat, dissipation of high-power mechanical and electrical equipment operation heat, heat release from coal rock oxidation, can also result in thermal damage [3,4]. Nowadays, the harm from thermal damage can be comparable yo disasters such as gas, water inrush, fire, dusts, and coal mining roof. Thus, the issue of thermal damage must be taken seriously [5].

The first step in the management of the mine thermal damage should be identifying the reasons for the high temperature and humidity of the mine. Then, the corresponding control measures can be designed according to the heat source. The control measures can be divided into three main

categories: (1) Heat removal, which is generally obtained through changing the development and deployment of roadway, ventilation and cooling, and optimizing the ventilation path, etc. (2) Heat insulation, which reduces the spread of stratum heat into the roadway. The most direct and effective method is to isolate the heat source. (3) Temperature reduction, which is obtained with the measures of artificially introducing cold media, such as ice, cooling water, and air conditioner in mines, etc. Compared with the traditional heat source research and control, the study of the temperature field of the tunnel's surrounding rock can predict the thermal environment of the roadway [6]. The temperature field of the surrounding rocks of the roadway with thermal insulation was simulated through the programming method, and the impact of the thermal insulation layer on the temperature field distribution was obtained [7]. Based on heat transfer and software analysis, the calculation formation for the convective heat transfer coefficient between air flow and surrounding rock was obtained, and the heat and mass transfer phenomena on the wall of mine roadway were numerically simulated [8,9]. Some researchers proposed a technology that coupled the insulation and support of roadway. In this proposed technology, the roadway insulation layer had certain support strength. When the strength of the insulation layer met the requirements, it can directly serve as support for the construction [10–12]. Based on this roadway insulation technology, researchers have performed a lot of studies on insulation materials. Using the developed thermal insulation materials, the traditional passive cooling method was changed to the active prevention method for thermal damage in which the heat source was shielded, the heat conduction was weakened, and the temperature of the roadway was reduced. The reduction of temperature in the roadway exhibited a negative exponential relationship with the thermal conductivity of the thermal insulation layer, which indicated that the thermal insulation with lower thermal conductivity had more obvious advantages [13–15].

Insulation materials refer to the materials or material composites with significant resistance to heat flow [16]. Insulation materials have been widely used in the construction industry. Among the insulation materials, basalt fiber, glass fiber and vitrified microbeads have been widely studied. Based on the existing researched, Yasir proposed to enhance the thermal performance using intumescent flame retardant coatings and investigated the synergistic effect of new fiber-reinforced materials (basalt fibers) on the thermal insulation performance of steel structures [17]. The thermal insulation performance was greatly improved by 2 wt % basalt fibers. Adding fibers instead of quartzite can improve the flexural and compressive strength of autoclaved aerated concrete, while carbon fiber reinforced autoclaved aerated concrete exhibited the best flexural and compressive strength [18]. Kizilkanat et al. compared the mechanical properties of reinforced concrete reinforced by basalt fiber and glass fiber. The results showed that as the fiber volume content increased, the tensile strength of basalt fiber reinforced concrete increased accordingly, while the tensile strength of glass fiber reinforced concrete had an initial increasing trend but remained unchanged after the volume content exceeded 0.5% [19]. Chen et al. studied the changes in chemical composition and mechanical properties of basalt fibers in the temperature range of 200~800 °C [20]. The results showed that in this temperature range, the fiber surface became smoother, the diameter of the monofilament became slightly smaller, and the mass was also decreased to some extent. The mass fractions of $SiO_2$ and $Al_2O_3$ both decreased, and the mass fraction of metal oxides increased. After the treatment at 200, 400, and 800 °C, the strength retention rates of the fibers were 98.3%, 64.6%, and 20%, respectively. The results fully indicated that basalt fiber had excellent high temperature resistance. Li et al. studied the variation law of the breaking strength of basalt fiber yarns after high-temperature treatment and the associated reactions [21]. The results showed that the breaking strength of the fibers first increased and then decreased with the increase of the temperature. This phenomenon was because when the temperature reached 200 °C, the distance between the two fibers decreased, which increased the frictional force of the fibers and increased the breaking strength. Gao established a mathematical model to predict the effective thermal conductivity of glass fiber boards [22]. Yang et al. used glass fiber to reinforce the phenolic foam, and found that the addition of glass fiber increased the ultimate oxygen index of the foam product by 8%, indicating that the flame retardancy of the phenolic foam was greatly improved [23–25]. Choe et al. proposed a

new microwave foaming method, which greatly improved the cell uniformity of phenolic foam [26]. At the same time, in order to improve the mechanical properties of phenolic foam, they introduced glass fibers and successfully prepared glass fiber-reinforced phenolic foam with excellent mechanical properties and heat resistance. Sayadi et al. used expanded polystyrene to replace vitrified microbeads in concrete, and studied the mechanical properties, thermal conductivity, fire resistance, resistivity, setting time, exothermic reaction and microstructural properties of the concrete [27]. Mahmoud et al. incorporated vitrified microbead powder into concrete to improve the compressive strength and durability of concrete [28]. After incorporating the vitrified microbeads into mortar, Zhang et al. tried to incorporate vitrified microbeads into concrete. They found that compared to ordinary concrete, the thermal conductivity of concrete mixed with vitrified microbeads was significantly decreased, while the strength did not have much decrease [29–32].

In general, using insulation materials to replace or cover traditional ordinary concrete can achieve the purpose of delaying the heat dissipation of surrounding rocks, reducing the air flow temperature, and reducing the cooling load of mining machinery. Diverse insulation materials were selected and the obtained test products were also different. Different products showed their own advantages, disadvantages, and applicability in the experiment and application. Although the obtained results in different tests varied from each other [33,34], overall, high-temperature mine thermal insulation materials had a strong applicability to the underground environment. In this paper, three materials, i.e., basalt fiber, glass fiber, and vitrified microbeads, were selected as the main raw materials to develop mine thermal insulation materials. By comparing the thermal conductivity and compressive properties of these three insulation materials, this paper provides a superior choice for the mining thermal insulation material.

## 2. Experimental Research on Mine Insulation Materials

### 2.1. Preparation of Experimental Materials

#### 2.1.1. Aggregate

Ceramsite was used as the aggregate, which is an inorganic material with the appearance of mostly round or oval spheres. Ceramsite has high impermeability and can retain in water and gas. It is mainly used to replace the natural river sand or mountain sand to prepare light aggregate concrete and lightweight mortar. Meanwhile, it is also a preferred fine aggregate for acid and heat resistant concrete. The specific parameters of ceramsite are shown in Table 1.

**Table 1.** Ceramsite basic parameter table.

| Test Material Name | Parameter Introduction | Physical Map |
| --- | --- | --- |
| ceramsite | colour: dark red; diameter: 3~5 mm; density: 980 kg/m$^3$. |  |

#### 2.1.2. Admixture

The selection of the admixture not only can replace part of the gelling material to reduce its amount and cost, but also can improve various properties of the mixed mixture and the hardened mixed material. This test involves three materials.

Basalt fiber is a new type of fiber that is resistant to heat, corrosion and chemicals [35]. Its specific parameters are shown in Table 2.

**Table 2.** Basalt fiber basic parameter table.

| Test Material Name | Parameter Introduction | Physical Map |
|---|---|---|
| basalt fiber | form: chopped strands;<br>basic performance: environmentally friendly and non-toxic;<br>monofilament diameter: 7~17 μm;<br>density: ≤3.0 g/cm$^3$;<br>fiber length: 0.3 or 0.6 cm;<br>elongation: 3.1%;<br>breaking strength: 3500~4300 MPa. | |

Glass fiber is made of natural ores as raw materials, mixed with substances such as soda ash and boric acid, and in the molten state, drawn or blown into extremely fine fibrous materials by external force [36]. The specific parameters are shown in Table 3.

**Table 3.** Glass fiber basic parameter table.

| Test Material Name | Parameter Introduction | Physical Map |
|---|---|---|
| glass fiber | form: chopped strands;<br>basic performance: white, environmentally friendly and non-toxic;<br>alkali content: <0.8;<br>monofilament diameter: 5~100 μm;<br>density: ≤1.0 g/cm$^3$;<br>fiber length: 0.5~2 cm;<br>elongation at break: ≤40%;<br>water absorption: ≤2%;<br>breaking strength: ≤450 MPa. | |

Vitrified microspheres is an inorganic lightweight aggregate glassy mineral material [37]. Specific parameters are shown in Table 4.

**Table 4.** Vitrified microspheres basic parameter table.

| Test Material Name | Parameter Introduction | Physical Map |
|---|---|---|
| vitrified microspheres | morphology: spherical particles;<br>basic requirements: fire resistance, high and low temperature resistance;<br>colour: white;<br>particle diameter: 0.15~1.5 mm;<br>porous hollow particles;<br>thermal conductivity: 0.036~0.054 W/(m·K);<br>bulk density: 50~200 kg/m$^3$;<br>scoring rate: 80%~95%;<br>water absorption: 20%~50%. | |

### 2.1.3. Additive

In this test, the cementitious material is cement. Its role is to form a slurry after stirring with water, which can harden in air or better in water and can firmly cement sand, stone and other materials together [38]. Superplasticizer is an admixture that can significantly reduce the amount of water mixed in concrete under the conditions that the slump of the concrete is basically the same [39]. The superplasticizer used in this experiment is a high-performance polycarboxylic acid superplasticizer. Specific parameters are shown in Table 5.

**Table 5.** Superplasticizer basic parameter table.

| Test Material Name | Performance | Physical Map |
|---|---|---|
| polycarboxylic acid superplasticizer | appearance: white fluid powder; outer packing density: 350~500 kg/m³; moisture content: ≤3%; loss on ignition: ≥ 85%;PH value: 7~8; Fineness (0.315mm sieve): ≥90%; $Cl^-$ content: ≤0.1; mortar water reduction rate: ≥25%. | |

### 2.2. Preparation of Test Specimens

Using an electronic balance to weigh cement, sand, ceramsite and water reducing agent, and set each inorganic insulation material at three levels. The inorganic materials in each group were weighed at 25%, 35%, and 45% of the cement weight Mix well; see Table 6 for specific material usage.

**Table 6.** Material dosage table.

| Inorganic Insulation Material (kg) | | Cement (kg) | Sand (kg) | Ceramsite (kg) | Water (kg) | Water Reducing Agent (g) |
|---|---|---|---|---|---|---|
| Material | Quantified Value | | | | | |
| basalt fiber | 0.30 (25%) | | | | | |
| glass fiber | 0.42 (35%) | 1.2 | 2.54 | 1.21 | 1.08 | 3.6 |
| vitrified beads | 0.54 (45%) | | | | 1.32 | |

Three test pieces were made for each level using a test die with the dimensions of 100 mm × 100 mm × 100 mm. After the mortar was completely stirred and discharged from the mixer, it was directly loaded into the mold, and then the mold was placed on a vibration table to ensure that the mortar in the mold was compacted. After 36 h, the mortar solidified to form a test piece. The test piece was removed from the mold using a demolding gun air pump and placed in a curing box containing a saturated solution of $Ca(OH)_2$ for 28 days. The indoor temperature was controlled at (20 ± 2) °C (GB/T 50080-2016) [40]. The specific test process is shown in Figure 1.

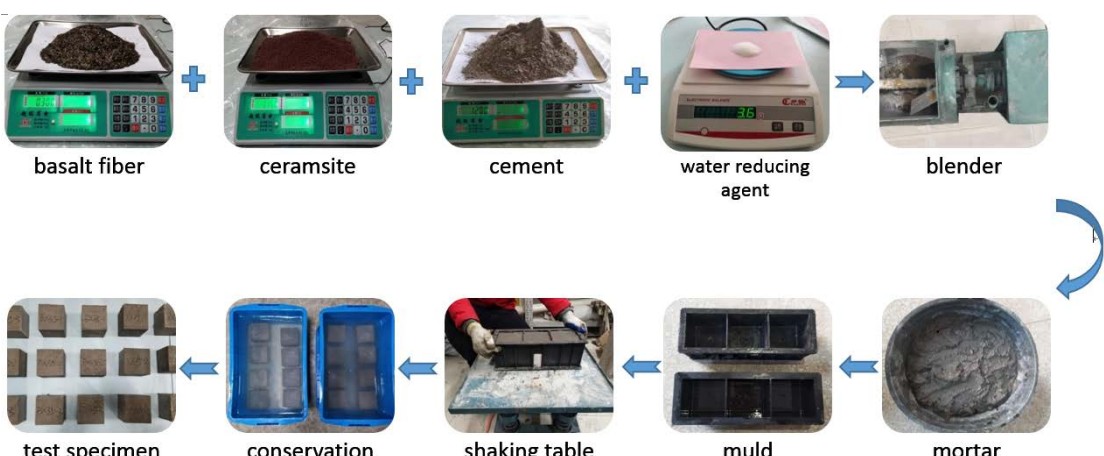

**Figure 1.** Flow chart of test specimen making.

### 2.3. Measurement of Thermal Conductivity

DRPL-I thermal conductivity tester (as shown in Figure 2a, Xiangtan Instrument Co., Ltd., Xiangtan, China) was used in the test. Each insulation material had 3 levels, and 3 test specimens

were prepared at each level. Each test specimen was measured with the thermal conductivity tester. At each level, if the relative error of the test results from three specimens at one level was within 10%, the average value was calculated as the thermal conductivity the material at this level; the test interface is shown in Figure 2b.

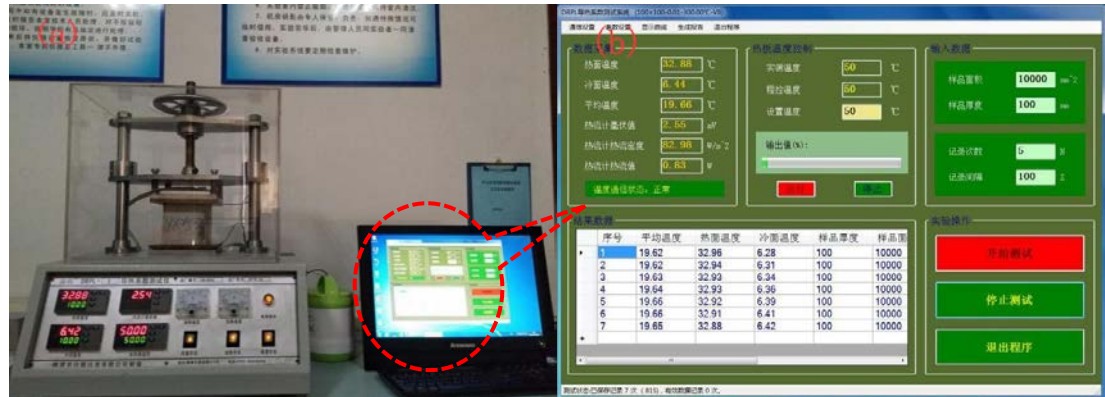

**Figure 2.** Measurement of thermal conductivity: (**a**) DRPL-I Thermal Conductivity Tester; (**b**) Test interface.

## 2.4. Measurement of Compressive Strength

Shimadzu AGX-250 electronic universal testing machine (as shown in Figure 3a, Shimadzu Corporation, Kyoto, Japan) was used in this test. Each insulation material had 3 levels, and 3 test specimens were prepared at each level. Each test specimen was measured with the electronic universal testing machine. At each level, if the relative error of the test results from three specimens at one level was within 10%, the average value was calculated as the compressive strength of this type of material at this level; the test interface is shown in Figure 3b.

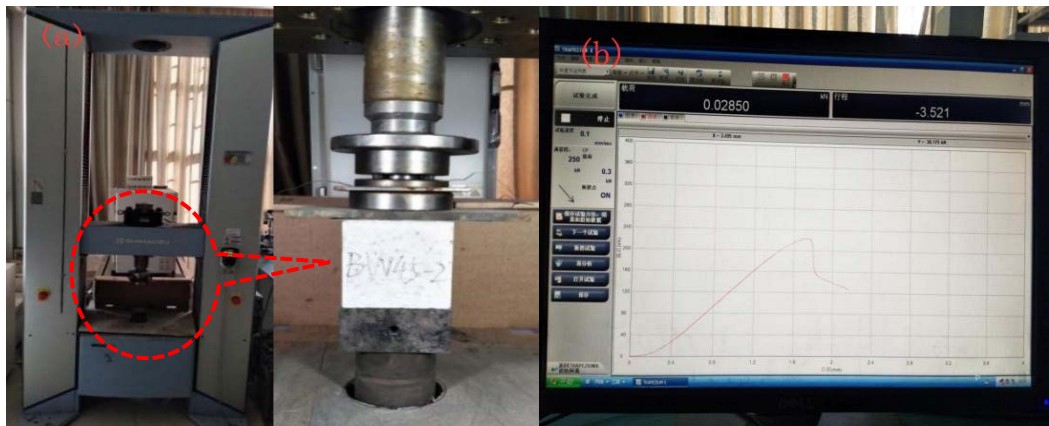

**Figure 3.** Process of determining compressive strength: (**a**) AGX-250 electronic universal testing machine; (**b**) test interface.

## 2.5. Numerical Simulation

COMSOL (5.4) was used to simulate the roadway with the length of 100 m long. The roadway was simplified into a cylinder. The outer diameter (heating circle radius) of the cylinder was $R$ = 32 m, the inner diameter $r$ was 2.15 m, and the average thickness of the shotcrete support inside the roadway was 0.15 m. Based on the above parameters, a simulation physical model was established and meshed, as shown in Figure 4. The number of grids was 131864. The numbers in the figure represent different grid qualities.

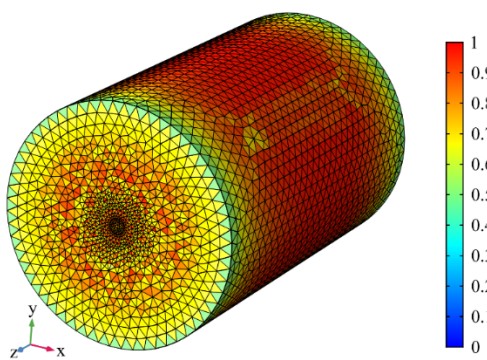

**Figure 4.** Physical model and mesh division.

In this numerical simulation, the model was solved by using the standard k-ε double equation turbulence model. The specific settings of the boundary conditions and initial parameters are shown in Tables 7 and 8.

**Table 7.** Boundary condition setting table.

| Place Name | Boundary Type | Code | Place Name | Boundary Type | Code |
|---|---|---|---|---|---|
| laneway wall | Wall | Wi | pressure outlet | Pressure-outlet | Po |
| surrounding rock surface | Wall | Wo | fluid outlet | Outflow | F |
| speed entry | Velocity-inlet | Vi | coupling boundary | Wall-shadow | Ws |

**Table 8.** Initial parameter settings table.

| Part Name | | Parameter Name | Model Parameters |
|---|---|---|---|
| roadway | | shape | cylinder |
| | | length/m | 100 m |
| | | outer diameter/m+temperature/°C | 32 + 40.7 |
| | | inside diameter/m+temperature/°C | 2.15 + 34.06 |
| shotcrete | | support thickness/m | 0.15 |
| shotcrete-I | Concrete | density/(kg/m$^3$) | 2400 |
| | | thermal conductivity/[W/(m·°C)] | 1.50 |
| merry | | speed/(m/s) | 2.5 |
| | | density/(kg/m$^3$) | 1.2 |
| surrounding rock | | thermal conductivity/[W/(m·°C)] | 2.10 |
| | | temperature coefficient/(m$^2$/s) | $0.843 \times 10^{-5}$ |
| others | | convection heat transfer coefficient/[W/(m$^2$·°C)] | 24.35 |

## 3. Analysis of Test Results

### 3.1. Measurement and Analysis of Thermal Conductivity

To obtain the lowest thermal conductivity of the test specimen, the moisture content of test specimen must be reduced to 0% [41]. Thus, we performed a dry process on the test specimen, during the drying process, the moisture content of each test specimen was measured and the average moisture content of each group of the specimens was calculated, as shown in Table 9.

**Table 9.** Moisture measurement data table.

| Specimen Number | Days | Average Moisture Content (%) | | |
|---|---|---|---|---|
| | | 1 d | 7 d | 28 d |
| BW | 25 | 5.75 | 2.76 | 0 |
| | 35 | 9.21 | 4.23 | 0 |
| | 45 | 6.96 | 3.12 | 0 |
| BX | 25 | 2.46 | 0.98 | 0 |
| | 35 | 2.46 | 0.99 | 0 |
| | 45 | 3.24 | 1.02 | 0 |
| XX | 25 | 2.55 | 1.02 | 0 |
| | 35 | 3.91 | 1.76 | 0 |
| | 45 | 5.83 | 2.61 | 0 |

In addition to the moisture content, the thermal conductivity of each group of specimens was measured and the average value was calculated. The detailed results are shown in Table 10.

**Table 10.** Thermal conductivity measurement data sheet.

| Specimen Number | Days | | Thermal Conductivity (Average Value) W/(K·m) | | | | | |
|---|---|---|---|---|---|---|---|---|
| | | | 1d | | 3d | | 28d | |
| BW | 25 | 1 | 0.4578 | | 0.4201 | | 0.3895 | |
| | | 2 | 0.4622 | (0.4567) | 0.4358 | (0.4279) | 0.3306 | (0.3667) |
| | | 3 | 0.4501 | | 0.4278 | | 0.3799 | |
| | 35 | 1 | 0.4101 | | 0.3632 | | 0.3174 | |
| | | 2 | 0.4098 | (0.4067) | 0.3541 | (0.3610) | 0.3035 | (0.3129) |
| | | 3 | 0.4002 | | 0.3658 | | 0.3179 | |
| | 45 | 1 | 0.3548 | | 0.3005 | | 0.2956 | |
| | | 2 | 0.3571 | (0.3560) | 0.3254 | (0.3146) | 0.2998 | (0.2928) |
| | | 3 | 0.3562 | | 0.3178 | | 0.2831 | |
| BX | 25 | 1 | 0.6211 | | 0.5795 | | 0.5552 | |
| | | 2 | 0.6405 | (0.6315) | 0.5788 | (0.5804) | 0.5369 | (0.5470) |
| | | 3 | 0.6330 | | 0.5829 | | 0.5489 | |
| | 35 | 1 | 0.5811 | | 0.5445 | | 0.5087 | |
| | | 2 | 0.5744 | (0.5753) | 0.5321 | (0.5424) | 0.5002 | (0.5073) |
| | | 3 | 0.5705 | | 0.5505 | | 0.5130 | |
| | 45 | 1 | 0.5341 | | 0.5056 | | 0.4852 | |
| | | 2 | 0.5228 | (0.5253) | 0.5000 | (0.5058) | 0.4698 | (0.4775) |
| | | 3 | 0.5191 | | 0.5118 | | 0.4775 | |
| XX | 25 | 1 | 0.4012 | | 0.3606 | | 0.3101 | |
| | | 2 | 0.3956 | (0.3924) | 0.3521 | (0.3565) | 0.3183 | (0.3212) |
| | | 3 | 0.3804 | | 0.3569 | | 0.3351 | |
| | 35 | 1 | 0.2951 | | 0.2216 | | 0.1779 | |
| | | 2 | 0.2874 | (0.2912) | 0.2347 | (0.2275) | 0.1822 | (0.1801) |
| | | 3 | 0.2912 | | 0.2261 | | 0.1802 | |
| | 45 | 1 | 0.2310 | | 0.1884 | | 0.1251 | |
| | | 2 | 0.2254 | (0.2302) | 0.1901 | (0.1870) | 0.1319 | (0.1323) |
| | | 3 | 0.2343 | | 0.1826 | | 0.1399 | |

As shown in Tables 9 and 10, the average moisture content and average thermal conductivity are considered as the moisture content and thermal conductivity of the material at that level. Test specimen XX 45 has the lowest thermal conductivity. A standard deviation analysis was performed on the measured thermal conductivity values to create Table 11.

**Table 11.** Standard deviation analysis comparison table of thermal conductivity of different thermal insulation materials.

| Material Content/%  Specimen Number | 25 | 35 | 45 |
|---|---|---|---|
| BW | 0.213% | 0.00668% | 0.00754% |
| BX | 0.229% | 0.00424% | 0.00593% |
| XX | 0.016% | 0.00460% | 0.00548% |

Based on the analysis of Table 11, when the material content was 25%, 35%, and 45%, the sample numbers with relatively small standard deviations were test specimen XX, test specimen BX, and test specimen XX. Therefore, the test results of the test specimen XX in the three materials have less fluctuations and the best stability.

Based on the analysis of the above table and Figures 5 and 6, it can be obtained that the moisture content of the test piece gradually decreased with time, and the thermal conductivity of the test piece gradually decreased accordingly. Thus, the water content was positively correlated with the thermal conductivity to a certain extent. According to the applicability requirements, mine thermal insulation materials should have a certain degree of water resistance and moisture resistance. Water has a far greater thermal conductivity than air, thus higher water content in the test specimen can inevitably lead to an increase in thermal conductivity. In addition, by comparing the results of the same materials at different levels, it can be found that as the content of the inorganic heat-insulating material in the test specimen increased, the thermal conductivity of the test specimen also gradually decreased.

By consulting the relevant standards(GB/T4272-2008) for determining thermal conductivity [42], in this test, we set the preliminary requirement for mine thermal insulation material of the thermal conductivity should not be higher than 0.30 W/(m·K). The measurement results show that the test specimen containing basalt fiber had the lowest thermal conductivity and basically met the design requirements. In addition, the test specimen at Level XX45 had the smallest the thermal conductivity, which was only 0.1323 W/(m·K) with a water content of 0. This result indicated that basalt fiber had a good thermal insulation effect. To a certain extent, as the basalt fiber content was higher, the thermal insulation effect was better.

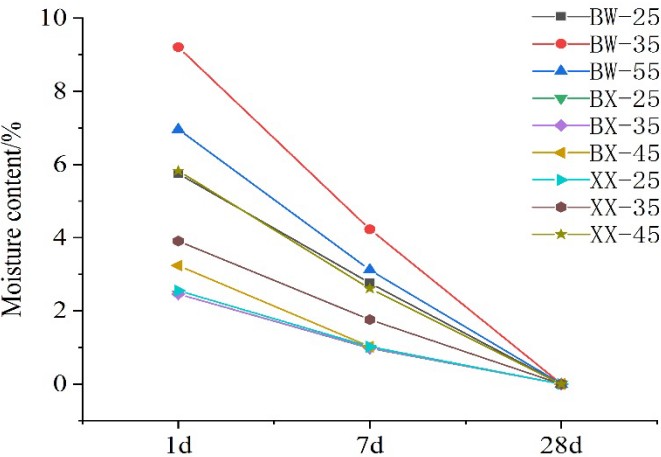

**Figure 5.** Relationship between test specimen moisture content and measurement time.

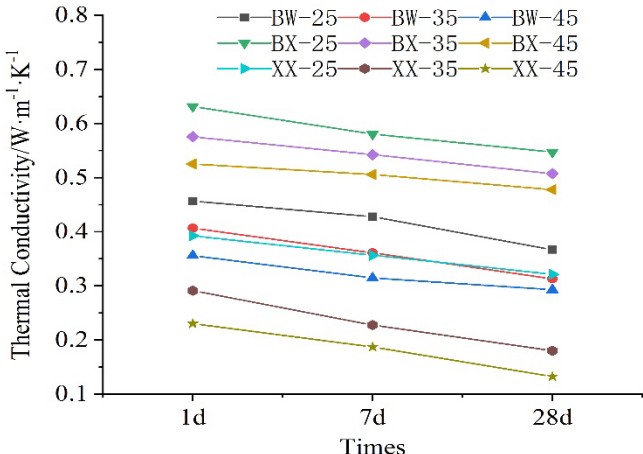

**Figure 6.** Change trend of thermal conductivity of test specimen with measurement time.

### 3.2. Measurement and Analysis of Compressive Strength

The compressive strength is one of the main indicators reflecting the quality of the material. In this test, the uniaxial compressive strength of the test specimens was measured. Uniaxial compressive strength is referred to as compressive strength, which is defined as the load per unit area the rock specimen can bear under unidirectional compression. The specific results are shown in Table 12.

**Table 12.** Summary table of compressive strength measurement data.

| Specimen Number | | Parameter | Compressive Strength (Average Value) MPa | |
|---|---|---|---|---|
| BW | 25 | 1 | 8.442110 | |
| | | 2 | 8.152187 | (8.46) |
| | | 3 | 8.790338 | |
| | 35 | 1 | 4.989604 | |
| | | 2 | 4.828739 | (5.13) |
| | | 3 | 5.561733 | |
| | 45 | 1 | 2.028973 | |
| | | 2 | 2.700337 | (2.48) |
| | | 3 | 2.701891 | |
| BX | 25 | 1 | 21.39526 | |
| | | 2 | 21.46985 | (21.31) |
| | | 3 | 21.05694 | |
| | 35 | 1 | 22.52147 | |
| | | 2 | 22.26835 | (22.27) |
| | | 3 | 22.01224 | |
| | 45 | 1 | 23.01549 | |
| | | 2 | 22.88358 | (22.99) |
| | | 3 | 23.08096 | |
| XX | 25 | 1 | 18.96541 | |
| | | 2 | 19.35963 | (19.34) |
| | | 3 | 19.68541 | |
| | 35 | 1 | 15.48659 | |
| | | 2 | 15.73184 | (15.46) |
| | | 3 | 15.15432 | |
| | 45 | 1 | 11.35646 | |
| | | 2 | 10.63288 | (10.98) |
| | | 3 | 10.94565 | |

As shown in Table 12, the average compressive strength is considered as the compressive strength of the material at that level. Test specimen BX45 has the strongest compression resistance. A standard deviation analysis was performed on the measured compressive strength values to obtain Table 13.

**Table 13.** Standard deviation analysis comparison table of compressive strength of different thermal insulation materials.

| Material Content/% Specimen Number | 25 | 35 | 45 |
|---|---|---|---|
| BW | 10.2% | 14.8% | 15.0% |
| BX | 4.80% | 6.4% | 10.0% |
| XX | 13.0% | 8.3% | 13.1% |

With regard to Table 13, compared with test specimen BW and XX under different material contents, the standard deviation of test specimen BX is the smallest, indicating that the test specimen BX has the smallest fluctuation and the best stability.

As can be seen in Figure 7, the compressive strength of the test specimen made of every material meets the requirements of the preliminary design (GB/T35056-2018), i.e., 2 Mpa [43]. Among the materials, the test specimens with the vitrified microspheres had relatively low compressive strength. The vitrified microspheres had small density, thus had high content in the test specimen. As a result, the test specimen had a higher pore rate, a lower density, and a lower strength. In addition, as the content of thermal insulation material increased, the compressive strength of BW and XX decreased, while the compressive strength of BX increased. From the analysis, the glass fiber used in the test had a low density and a very small diameter (5 to 100 μm), which was much smaller than the diameter of other materials for the test specimens. As a result, in the synthesis process of the test specimens, the glass fiber not only had a certain thermal insulation effect but also played a strong role in polymerization and coagulation. To a certain extent, as the content of glass fiber was higher, the polymerization and coagulation effect was more, and the compressive strength of the obtained test specimens was higher. The test specimen with basalt fiber had a lower compressive strength than the test specimen with glass fiber. However, the compressive strength of the test specimen with basalt fiber was still 5 to 10 times as high as the design standard of the compressive strength. Thus, the compression performance of the test specimen with basalt fiber was very good.

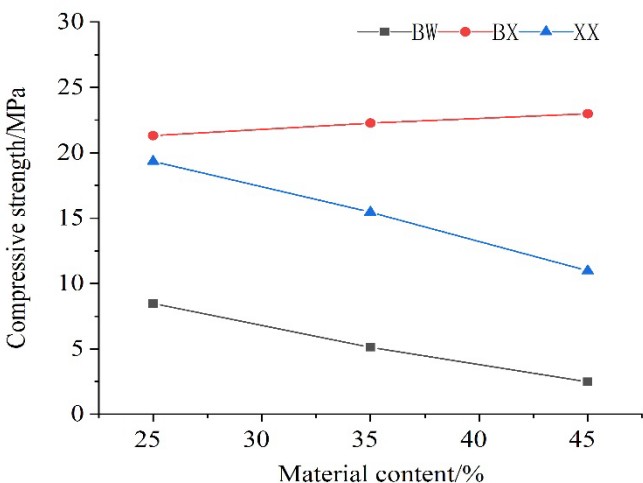

**Figure 7.** Variation trend of compressive strength with the amount of thermal insulation material.

In summary, it can be simplified with the comparison table of the performance parameters of the test specimen admixture materials shown in Table 14 below.

**Table 14.** Material performance parameter comparison table.

| Parameter  Performance Parameter   Material | Thermal Conductivity | Compressive Strength |
|---|---|---|
| BW | moderate | softest |
| XX | fastest | moderate |
| BX | slowest | hardest |

It can be seen that the thermal conductivity of the test piece XX45 blended with basalt fiber is relatively lowest, and the compressive strength meets the requirements and the selection conditions of the mine thermal insulation material.

### 3.3. Comparative Analysis of Simulation Verification

From the thermal conductivity and compressive strength, it can be seen that the performance of the test specimen XX containing basalt fiber is more suitable as a thermal insulation material for mining. Therefore, the test specimen XX45 was used in simulation verification and the performance of XX45 was compared with ordinary concrete materials. The parameters of the test specimen XX45 are summarized in Table 15.

**Table 15.** Parameter table of test specimen XX45.

| Part Name | | Parameter Name | Model Parameters |
|---|---|---|---|
| shotcrete -II | XX45 | density/(kg/m$^3$) | 2610 |
| | | thermal conductivity/[W/(m·°C)] | 0.1323 |

The results are shown in Figure 8a,b, Figure 9a,b, and Figure 10a,b.

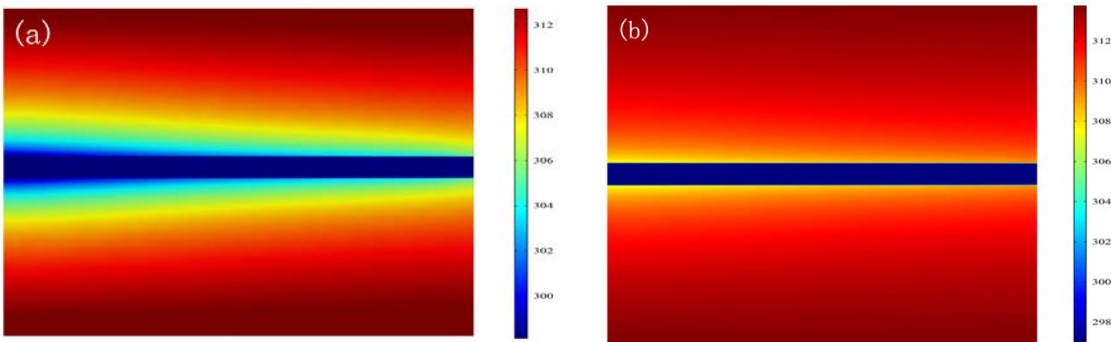

**Figure 8.** Comparison chart of thermal insulation effect of supporting materials (cross section): (**a**) cloud diagram of the temperature field of the concrete support roadway; (**b**) cloud diagram of the temperature field of the roadway supported by the test specimen XX45 insulation material.

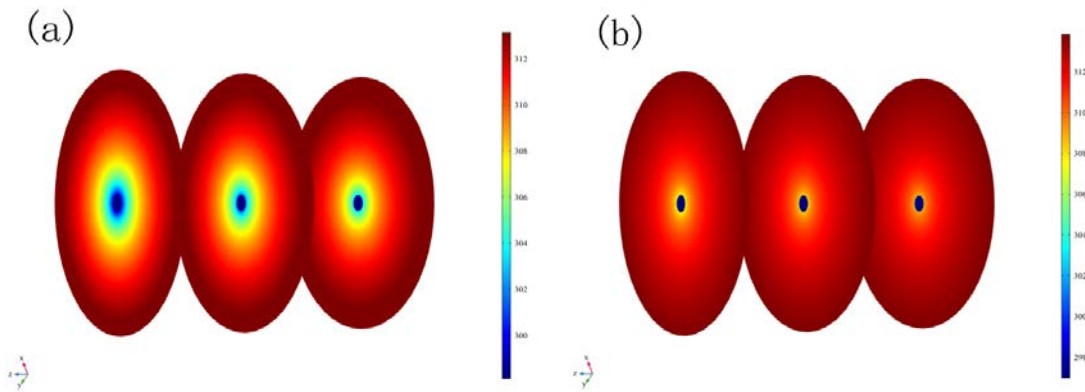

**Figure 9.** Comparison of thermal insulation effect of supporting materials (longitudinal section): (**a**) cloud map of the temperature field of the concrete support roadway; (**b**) cloud diagram of temperature field of test specimen XX45 insulation support roadway.

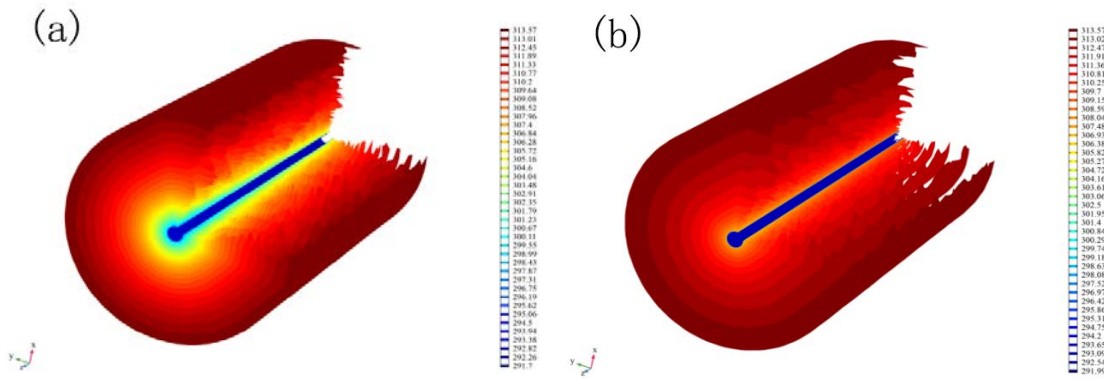

**Figure 10.** Temperature isosurface: (**a**) temperature isosurface of concrete support roadway; (**b**) temperature isosurface of roadway supported by test specimen XX45 insulation material.

From each type of support material, the thermal conditions on the model outer boundary are the same, and the temperature of the airflow is the lowest at the air inlet. From Figure 8a, the temperature of the surrounding rocks shows a clear gradual trend. Due to the higher thermal conductivity of the concrete supporting material, the heat dissipation of the surrounding rock had a great influence on the air flow, and more cooling capacity was used to balance the heat dissipation of the surrounding rock. Correspondingly, in a unit time, more heat was emitted by more surrounding rocks and entered the roadway through the concrete supporting material, causing the temperature of the air flow to rise. As the air flow temperature gradually increased, its regulating effect on surrounding rocks became smaller and smaller, and its influencing scope was more limited. The overall effect of heat exchange between air flow and surrounding rocks shown in Figure 8b was much less obvious than that shown in Figure 8a. This is because the thermal conductivity of the test specimen XX45 mine thermal insulation material is relatively small, which greatly prevents the heat of the surrounding rocks from being dissipated into the roadway. As a result, the utilization efficiency of the cooling airflow was improved and the thermal insulation effect was significantly enhanced.

From Figure 9a, the effect of air flow temperature on the surrounding rock of the roadway became smaller and smaller, which was indicated by the smaller circular area of the same color in the figure. In Figure 9b, although the change is not obvious, a similar trend was also shown. It can be seen from the different figures that the test specimen XX45 mine thermal insulation material had a good thermal insulation performance. At the same location, the temperature in Figure 10b is significantly lower than that shown in Figure 10a.

Then, the model was quantized. A straight line parallel to the cross-section and passing through the center of the section was drawn at the entrance, center, and exit of the model, as shown in Figure 11.

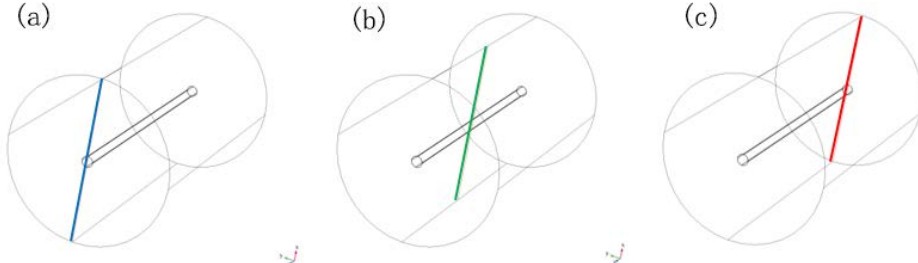

**Figure 11.** Standard linear position map for material temperature statistics: (**a**) entrance; (**b**) center; (**c**) exit.

We exported the three sections through the temperature value extracted by the custom domain probes in COMSOL simulation software to draw a two-dimensional plot, as shown in Figure 12.

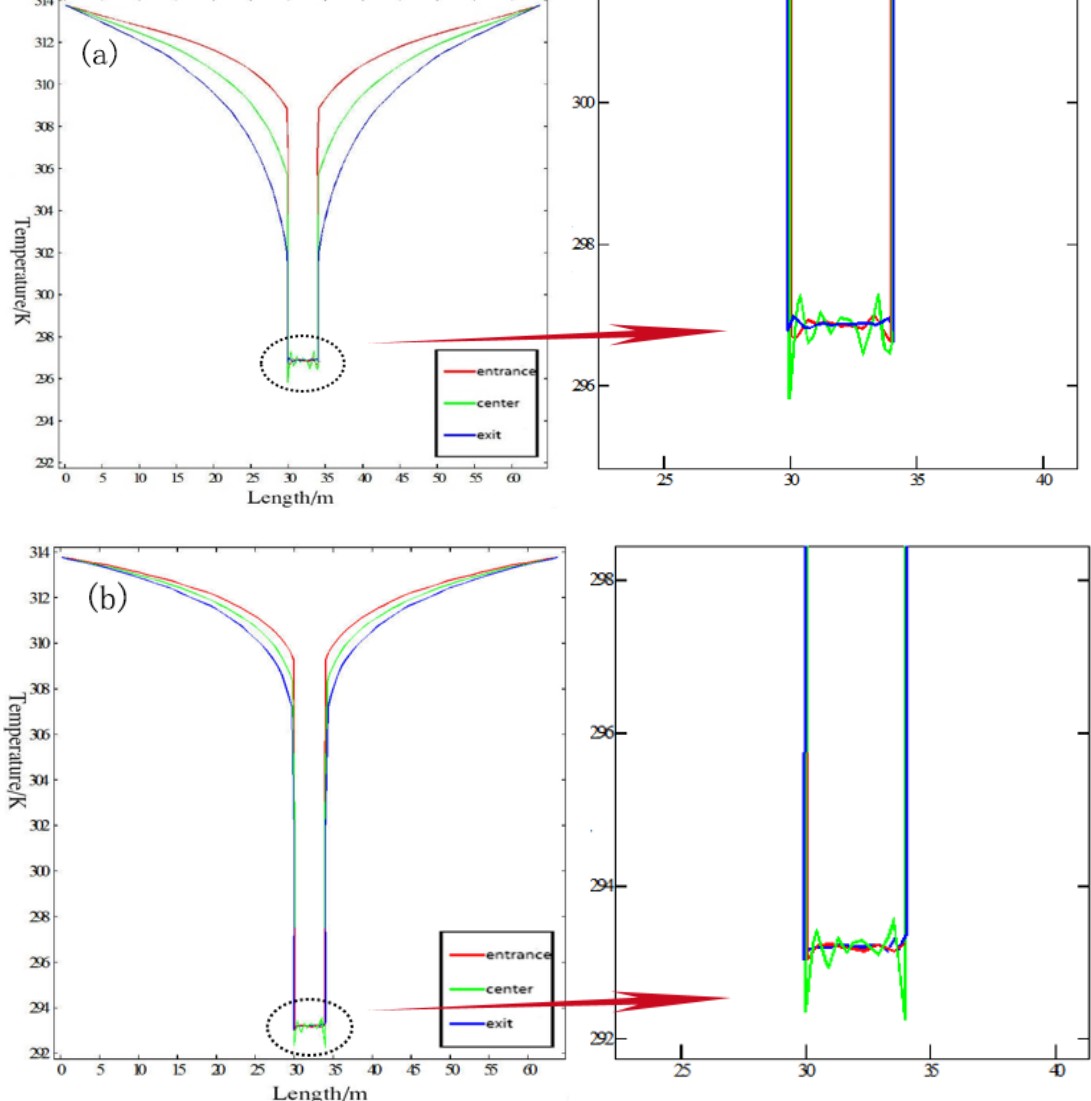

**Figure 12.** Temperature statistics of materials along standard straight lines: (**a**) temperature chart of concrete material; (**b**) temperature chart of test specimen XX45.

Due to the symmetry of the model, in the line graph in Figure 12a,b, the temperature was the lowest at the inlet and the highest at the outlet. At a length of 0 (outer surface of the model), the temperature was approximately 314 K, i.e., the initial temperature of 313.7 K. Take the temperature at the center (Figure 11b) as an example; in Figure 12a, the initial temperature of the surrounding rock was transmitted inward along the standard straight line until the temperature at the wall of the roadway (31.5 m) dropped to about 305.5 K. Under the effect of the concrete material and the airflow in the roadway, the temperature dropped straight to about 297 K. In Figure 12b, the temperature at the wall surface of the roadway (31.5 m) was dropped to about 308.5 K. Due to the heat barrier effect by the XX45 mine insulation material, the temperature dropped straight to about 293 K. Because the influence of air flow on the temperature of the surrounding rock in the roadway was affected by the heat barrier effects of the two materials, the temperature difference on the wall surface of the surrounding rock was about 3 K and the temperature difference of the air flow in the roadway was about 4 K. However, the temperature difference on the surrounding rock wall surface was about 5 K at the entrance and about 1 K at the exit. The decrease in the temperature difference was due to the temperature rise caused by the heat exchange between the air flow and the surrounding rock. The appearance of the temperature difference also verified that the XX45 mine thermal insulation material had good thermal insulation performance.

## 4. Conclusions

By processing and analyzing the test results, we can conclude the following: (1) According to the measured data of the thermal conductivity of the test specimens, we found that test specimen XX45 with basalt fiber had the smallest thermal conductivity, which was only 0.1323 W/(m·K) when the moisture content was 0 (after drying). Compared with traditional concrete insulation materials (1.50 W/(m·K)), the thermal conductivity of test specimen XX45 was reduced by about 1/12. This indicates that basalt fiber had a good thermal insulation effect. In addition, to a certain extent, as the content of basalt fiber became higher, the thermal insulation effect was better. (2) According to the measured data of the compressive strengths of different test specimens, we can find that the test specimen XX45 with basalt fiber had moderate compressive strength, which was 10.98 Mpa. Although the compressive strength is not the largest among the test specimens, test specimen XX45 reached the design standard of compressive strength. However, compared with the compressive capacity of traditional concrete, it was only 1/3, and its compressive capacity still needs to be improved. (3) At least, based on the statistical analysis on the temperature at the entrance, center, and exit of the model, the temperature difference between two materials was about 3 K on the wall surface at the center and about 4K in the air flow in the roadway. The analysis results confirm that the test specimen XX45 with basalt fiber had better thermal insulation performance than traditional concrete materials.

**Author Contributions:** C.H. and S.X. conducted the main work and wrote the paper; L.Z. completed basic experiments; S.L. completed the data processing and analysis; X.Z. designed the numerical simulation experiment and operation; they all provided insightful suggestions and revised the paper. All authors have read and agreed to the published version of the manuscript.

**Funding:** This work has been funded by the National Natural Science Foundation of China, grant numbers 51774197. Focus on Research and Development Plan in Shandong Province, Grant number 2018GSF116012, the Natural Science Foundation of Shandong Province, Grant number ZR2016EEP02.

**Conflicts of Interest:** The authors declare no conflict of interest.

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
