# Peer review of "Foundation Research on Physicochemical Properties of Mine Insulation Materials"

_coatings, doi:10.3390/coatings10040355_

Round 1
Reviewer 1 Report
The paper analyzes the physicochemical properties of three types of insulation materials used in coal mines: basalt fiber, glass fiber and vitrified microbeads.
The manuscript presents a clear structure and easy to follow way.
However, the properties of the new materials should be presented in comparison with the insulation materials used currently. The advantages and disadvantages should be highlighted in Conclusion section.
Editing corrections are also required according to the Coatings template.
Reviewer 2 Report
“Foundation Research on Physicochemical Properties of Mine Insulation Materials” investigates on three types of mineral insulation materials,
prepared using basalt fiber, glass fiber, vitrified microbeads in combination with cement, sand, high-strength ceramsite, water. The thermal conductivity and compressive strength of the prepared specimens were assessed. Results were implemented in a mine roadway case by a simulation.
Revision or comment
The work is interesting and the experimentation seems adequate but results are not clearly presented both for some important conceptual missing (e.g. the uncertainties) and for editing missing. Simulation gives strength to the results showing the impact of employed tested material with best performances in a roadway of a mine. However, the last part of the paper presents a “statistical” analysis on the temperature that is not clear. Some improvements could enhance the paper.
Line 178: It could be useful to know the number of mesh elements, (domains, boundaries, edges).
Line 194: The standard deviations of the reported measurements are missing. They are useful (essential) to understand the difference among the levels, of the thermal conductivity for the different insulation materials used.
Line 224: Same consideration on standard deviations missing.
Line 210, 231 : How are the preliminary requirement for mine thermal insulation material and compressive strength identified?
Line 291: It is not clear which statistical analysis has been performed. How many values has been considered? what method?
Line 299: Please, use always the same temperature scale (Kelvin or Celsius).
Authors are invited to inspect the paper for minor revisions like the follow:
Table 2 and 3 have different format in comparison to table 4 and 5 that are clearer.
Line 168,177: figure 2 and 3, 4 and 5 could be indicated as same figure 2 a) and b), 3 a) and b).
Line 185: In figure 6 could be inserted the physical model (as seen in the geometry) and only in one mesh division (the left one in the current version).
Line 260-262: figure caption is not clear and probably a detail is missing about a material of figure a). It could be better to write figure 11…., a)…,b)… and not a) and b) before the figure number.
The same for figure 12.
Line 302-303: Figure caption missing.
Round 2
Reviewer 2 Report
The revised paper is now clearer than the previous version.
Minor revision:
Fig.4 Colour scale legend is missing.
Fig. 12 Use for the figure caption the same format of the previous figures.
Figure 12..............a),............b)